# Total enzymatic synthesis of cis-α-irone from a simple carbon source

Xixian Chen [1] ✉, Rehka T[1], Jérémy Esque [2], Congqiang Zhang [1], Sudha Shukal[1], Chin Chin Lim[1], Leonard Ong [1], Derek Smith [1] & Isabelle André [2] ✉

Metabolic engineering has become an attractive method for the efficient production of natural products. However, one important pre-requisite is to establish the biosynthetic pathways. Many commercially interesting molecules cannot be biosynthesized as their native biochemical pathways are not fully elucidated. Cis-α-irone, a top-end perfumery molecule, is an example. Retro-biosynthetic pathway design by employing promiscuous enzymes provides an alternative solution to this challenge. In this work, we design a synthetic pathway to produce cis-α-irone with a promiscuous methyltransferase (pMT). Using structure-guided enzyme engineering strategies, we improve pMT activity and specificity towards cis-α-irone by >10,000-fold and >1000-fold, respectively. By incorporating the optimized methyltransferase into our engineered microbial cells, ~86 mg l$^{-1}$ cis-α-irone is produced from glucose in a 5 l bioreactor. Our work illustrates that integrated retrobiosynthetic pathway design and enzyme engineering can offer opportunities to expand the scope of natural molecules that can be biosynthesized.

Genetically engineered microorganisms have been rapidly developed as the workhorse to produce essential molecules at unprecedented yield[1–4]. Technological advancement has enabled the simultaneous optimization of multiple genes in a single microbial strain. To date, multiple genes can be rapidly assembled and transformed into microbial strains to probe the collective functions of a set of genes. In this manner, many complex and valuable natural products can be produced. However, one important prerequisite is the availability of prior knowledge about the product's biosynthetic pathway. Unfortunately for many industrially important molecules, their biosynthetic pathways are often incomplete with many missing enzymatic reactions.

Cis-α-irone, the principal component of orris root oil, is a good example. Traditionally used in Queen's water, orris oil commands a hefty price in the fragrance industry due to its lengthy and inefficient manufacturing process. With a production period spanning over three years, approximately 30−70 mg of natural irone can be produced from

1 kg of fresh iris rhizome. In the 1990s, Gil et al. patented an enzymatic process to degrade irone precursors from orris roots with lipoxidase or peroxidase[5]. The method improved the production rate but was still reliant on orris plant materials. Furthermore, different species of irises give rise to divergent odour perceptions because the composition of irone isomers are non-identical[6,7]. There are 10 different regio- and stereoisomers of irone[8], and olfactory studies indicate that cis-α-irones, but not trans-α-irones, are aromatic[6,8]. A semi-synthetic method exists to produce irones (Irone Alpha® by Givaudan) from methyl psi-ionone, a non-natural substrate. Irone Alpha® is a racemic mixture containing 42% cis-α-irones ((1 S,5 R)-cis-α-irone (1S5R) and (1 R,5 S)-cis-α-irone (1 R5S)), 53% trans-α-irones ((1R,5 R)-trans-α-irone (1R5R) and (1 S,5 S)-trans-α-irone (1S5S)) and 5% β-irones[8]. Lipase-mediated synthesis can resolve the racemic mixture of Irone Alpha®[6]. However, the process involves >5 steps of oxidation and reduction, affecting the purity of the final product. Total enzymatic synthesis of cis-α-irone through microbial fermentation could potentially eliminate

[1]Singapore Institute of Food and Biotechnology Innovation (SIFBI), Agency for Science, Technology and Research (A*STAR), Singapore. 31 Biopolis Way, Level 6 Nanos building, Singapore 138669, Singapore. [2]Toulouse Biotechnology Institute, TBI, Université de Toulouse, CNRS, INRAE, INSA, Toulouse, France. 135, avenue de Rangueil, F-31077 Toulouse, Cedex 04, France. ✉e-mail: Xixian_chen@sifbi.a-star.edu.sg; isabelle.andre@insa-toulouse.fr

the reliance on orris rhizomes and shorten the production process, but this method is not yet feasible because there are multiple unknown enzymes along the irone biosynthetic pathway. A radio-labelling study in the 1980s demonstrated that irones are oxidative degradation products of iridal triterpenoids[9]. The study proposed that the irone moiety is formed by a bifunctional methyltransferase / cyclase (bMTC). However, this plant enzyme is yet to be identified.

To circumvent this challenge, in this work, we demonstrate an artificial enzymatic synthetic route to produce cis-α-irone from glucose by identifying and optimizing a promiscuous bifunctional methyltransferase / cyclase enzyme (pMT). With structure-guided rational design, we improve the activity and selectivity of pMT towards cis-α-irone by >10,000-fold and >1000-fold, respectively. About 91.1 mg l⁻¹ cis- α-irone and 42.9 mg l⁻¹ β-irone are produced by one-step biotransformation of synthetic psi-ionone. By incorporating the engineered pMT enzyme in the psi-ionone-producing microbe, we achieve ~ 86.0 mg l⁻¹ cis- α-irone and 35.6 mg l⁻¹ β-irone production from glucose in a fed-batch process. This work paves the way towards developing an orris-independent route to biosynthesize cis-α-irone. The integrated synthetic pathway design and rational enzyme engineering approach is a promising method to produce natural products whose native biosynthetic pathways are yet to be discovered.

## Results and discussion
### Synthetic pathway design and validation
From the chemical synthesis route of Irone Alpha® (chemical route, Fig. 1), we postulated that psi-ionone could be one of the precursors to irone. Moreover, based on the previous radio-labelling study[9] (native biosynthesis route, Fig. 1), we hypothesized that a promiscuous bMTC

enzyme might directly convert psi-ionone to irone in a single step instead of via two sequential methylation and cyclization steps (artificial biosynthesis route, Fig. 1). Psi-ionone can be biosynthesized from lycopene, which can be produced from glucose using metabolically engineered cells. Thus, retrosynthetically, irone can be produced from simple carbon such as glucose in an orris-independent manner. Moreover, two molecules of psi-ionone can be formed from symmetric lycopene by carotenoid cleavage dioxygenase (CCD1)[10,11] and converted to two molecules of irone, whereas only one molecule of irone can be formed from asymmetric iridal (Fig. 1). Theoretically, the artificial route has a higher carbon yield than that for the native biosynthetic route from their respective terpenoid precursors (65% vs 46.7%, Fig. 1). Serendipitously, a bifunctional methyltransferase and cyclase (TleD or pMT1 in this study) was recently discovered from Streptomyces[12] and structurally elucidated (PDB id: 5GM1 (substrate-free), 5GM2 (substrate-bound))[13]. TleD methylates the terpene moiety of teleocidin A1 and cyclizes it to form a 6-member ring. Along with TleD, we also tested another three methyltransferases with known structures (Supplementary Table 1): cyclopropane mycolic acid synthase 1 from Mycobacterium tuberculosis (Mt.CMAS)[14], geranyl diphosphate 2-C-methyltransferase from Streptomyces lasalocidi (Sl.GdpMT)[15], and a methyltransferase-like protein from Saccharopolyspora spinosa (Ss.SpnF)[16]. By incubating cell extracts overexpressing the four respective methyltransferases with psi-ionone, only reaction with pMT1 gave rise to detectable levels of trans-α-irones (84%) and cis-α-irones (16%) (Fig. 2a and b).

To identify a potentially more active enzyme, the pMT1 sequence was mined using the Basic Local Alignment Search Tool (BLAST) against the non-redundant protein sequence database[17]. Two sequences were

**Fig. 1 | Schematic representations of the different pathways to produce irone.** The chemical route uses acid to catalyse cyclization of methyl-3-psi-ionone, which is not a known natural product. Chemical synthesis produces a racemic mixture containing (1R, 5R)-trans-α-irone (1R5R), (1S, 5S)-trans-α-irone (1S5S), (1S, 5R)-cis-α-irone (1S5R), (1R, 5S)-cis-α-irone (1R5S), (5R)-β-irone and (5S)-β-irone. The native biosynthesis route was proposed based on a radiolabelling study[9], and a bifunctional methyltransferase and cyclase (bMTC) was hypothesized to convert iridal directly to cycloiridal. Cycloiridal is asymmetric, so only one molecule of α-irone is produced from the C30 precursor. The resulting carbon yield is 46.7%. The artificial biosynthesis route was designed by combining both the chemical and biological synthesis knowledge, namely, to use a promiscuous bMTC to convert psi-ionone into α-irone. Two molecules of psi-ionone molecules are readily formed from the symmetric C40 lycopene; hence, two molecules of α-irone are expected to be formed from the C40 precursor. The resulting carbon yield is 65%, which is higher than the native biosynthesis route. The abbreviations are as follows. SAM: S-adenosylmethionine. SAH, S-adenosylhomocysteine; CCD1, carotenoid cleavage dioxygenase 1.

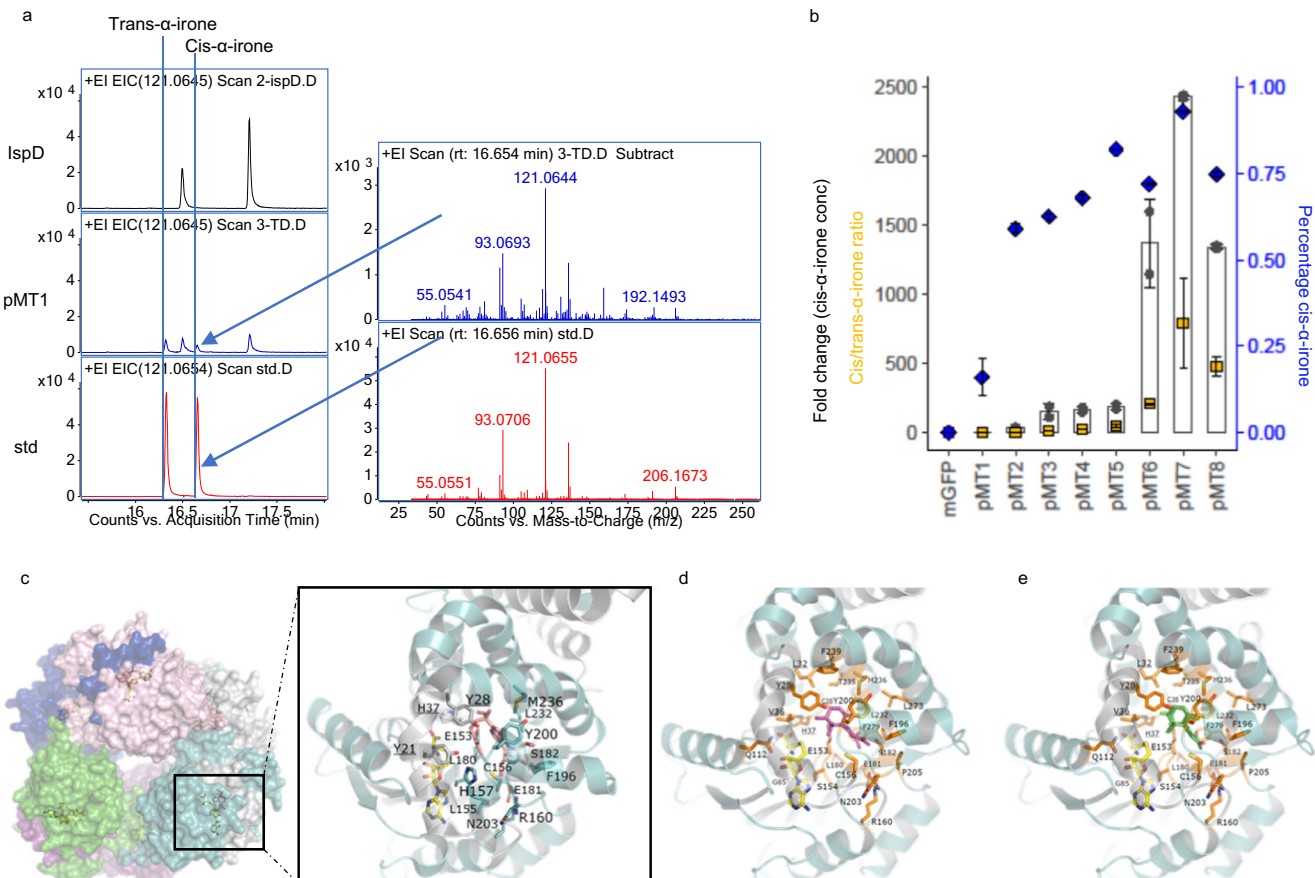

**Fig. 2 | Promiscuous methyltransferase catalyses psi-ionone to α-irone. a** Cell lysates overexpressing IspD (negative control) or pMT1 were used to assay against psi-ionone. The reaction mixtures were incubated at 28 °C for 2 days and subjected to headspace solid-phase microextraction coupled to gas chromatography-mass spectrometry (HS-SPME-GCMS) analysis. Methylated products that correspond to trans-α-irone and cis-α-irone were detected in the reaction containing pMT1. The retention time and mass spectrum of the irones detected are the same as the synthetic chemical standard (std) from Sigma Aldrich. **b** Cis-α-irone, trans-α-irone and β-irone were quantified for each pMT mutant by HS-SPME-GCMS against an external standard curve. The bar chart represents the average fold change in cis-α-irone concentration compared to pMT1. The ratio between cis-α-irone to trans-α-irone concentration was calculated and represented as orange squares. The

percentage of cis-α-irone in the irone mixture was calculated by dividing cis-α-irone concentration by the total irone concentration and represented by blue diamonds. The average and the standard deviation (s.d.) of two biologically independent experiments are shown. **c** Crystallographic structure of hexameric pMT1 in complex with a cofactor *S*-adenosylhomocysteine (yellow sticks) and the substrate teleocidin A1 (pink sticks) (PDB id: 5GM2). Enlarged view of amino acid residues lining the active site is shown. Three-dimensional models of pMT1 with **d** cis-α-irone (1S5R and 1R5S in magenta) **e** trans-α-irone (1S5S and 1R5R in green) are shown. pMT1 is shown as cartoon with one chain in cyan and the second one in grey. The 24 amino acid positions selected for site mutagenesis are displayed as orange sticks. Labels of residues from the second chain are underlined. Source data are provided as a Source Data file. The pdb files are available as Supplementary data 1.

identified to have >60% sequence identity to pMT1 (Supplementary Fig. 1). Both enzymes were recombinantly produced in *Escherichia coli* (*E. coli*) and tested. Only methyltransferase from *Streptomyces albireticuli* (SaMT, ~68% sequence identity to pMT1) converted psi-ionone to predominantly trans-α-irone. The other ~66% homologous enzyme from *Streptomyces clavuligerus* (ScMT) was not active towards psi-ionone (Supplementary Fig. 1). Since the crystallographic structure of pMT1 was available, we decided to engineer pMT1 to further improve its catalytic efficiency.

**Structure-guided enzyme engineering to improve pMT activity**
In solution and in X-ray structure, pMT1 shows a hexameric assembly whose minimal functional unit is a dimer[13]. pMT1 displays a unique active site formed by amino acid residues from two distinct monomeric chains of the hexamer, with the N-terminus of one monomer covering the catalytic cavity of the other monomer (Fig. 2c). Using the structure of pMT1 in complex with the cofactor *S*-adenosylhomocysteine (SAH) and the substrate teleocidin A1 (PDB id: 5GM2) as a template, we modelled pMT1 bound to each of the four isomers of α-irone with the aim of identifying amino acid residues that could be

critical to increase cis-α-irone or decrease trans-α-irone production (Fig. 2d, e). Visual inspection of the pMT1: α-irone complexes enabled us to identify 24 amino acid residues lining the substrate- and cofactor-binding pocket. Site-directed mutagenesis of the 24 residues was carried out with degenerative primers. Subsequently, we pooled the mutants of each site together, and examined if the activity or product-selectivity of the pooled mutants were increased (Supplementary Fig. 2). This reduced the number of screening reactions to 24, which was manageable when a high-throughput enzymatic assay was not available. Among them, site-saturated mutants of G85 were also included. This is a key residue involved in SAM co-factor binding. Mutating this glycine to the other 19 amino acids significantly reduces the methyltransferase activity. Hence, we reasoned that the activity of the pooled G85 mutants could be an appropriate baseline to assess the impact of the mutation on the activity of pMT: if the pooled mutants have lower or similar activities as G85 mutants, the corresponding residues are probably essential for methyltransferase activity and should not be mutated; otherwise, the corresponding residues are potential targets for further analysis to identify the specific beneficial mutation.

**Table 1 | Kinetics parameters of pMT7 and pMT10**

|  | Substrate | $K_m$ (µM) | $k_{cat}$ (x $10^3$ $h^{-1}$) | $k_{cat}/K_m$ ($M^{-1}s^{-1}$) | $IC_{50}$ SAH (µM) |
|---|---|---|---|---|---|
| pMT7 | Psi-ionone | 24.9 ± 3.7 | 2.2 ± 0.11 | 0.024 | |
| | SAM | 53.6 ± 1.8 | 2.6 ± 0.025 | 0.013 | 4.9 |
| pMT10 | Psi-ionone | 18.1 ± 0.03 | 19.9 ± 0.1 | 0.30 | |
| | SAM | 31.4 ± 2.4 | 25.8 ± 0.47 | 0.23 | 3.1 |

Among the 24 pooled reactions, seven positions displayed at least a two-fold improvement in cis-α-irone production compared to the baseline activity (i.e., G85 mutation), and three out of the seven residues, namely Y200, L180 and R160, increased cis-α-irone production compared to pMT1 (Supplementary Fig. 2). Moreover, more than half of the mutated residues showed altered product selectivity, increasing the cis-to-trans-α-irone ratio compared to pMT1. Among them, S182, which showed the highest cis-to-trans-α-irone ratio (Supplementary Fig. 2), was the most promising residue to be mutated.

Subsequently, site-saturated mutagenesis of Y200 was carried out. A Y200F (pMT2) mutation improved the soluble expression of the methyltransferase and its total turnover number (TTN) by ~59.7-fold (Supplementary Fig. 3). More importantly, the cis-isomer content increased to ~59.1% (Fig. 2b). It is notable that a Y200L mutation, a naturally occurring residue in SaMT, improved trans-α-irone production (Supplementary Fig. 1b), suggesting that Y200 plays a role in controlling the stereoselectivity of the products.

Next, we performed site-saturation mutagenesis on L180 and S182 using pMT2 as the new template. A L180 mutation to alanine resulted in a marginal increase in cis-α-irone production. One mutant, S182E (pMT3), improved TTN and cis-α-irone production by ~173.4- and ~149.4-fold, respectively, as compared to pMT1 and increased the cis-α-irone content to ~62.6% (Fig. 2b and Supplementary Fig. 3b). Structure and predicted pKa analysis revealed that E182 (predicted pKa = 7.2) is mostly protonated under neutral pH, thus it may establish a polar contact with the ketone moiety of psi-ionone or irone. This polar interaction could be important for enzymatic activity. Indeed, mutating S182 to aspartate did not elicit the same enzymatic activity, as the distance between the carboxylate side chain and the ketone moiety probably became too long for any polar contact. Moreover, when the adjacent F196 was further mutated to arginine, creating a salt bridge between E182 and R196, the methyltransferase activity towards psi-ionone was nearly abolished (Supplementary Fig. 4).

To further improve the enzymatic activity and selectivity of the promising mutant (pMT3), molecular dynamics (MD) simulations were conducted using the modelled complexes of the four isomers of α-irone and pMT1 (or pMT3) to identify amino acid residues that could be critical to improve the binding of cis-α-irone or decrease the binding of trans-α-irone. Visual inspection and free energy calculations were performed. Interestingly, free energy calculations showed a change of the free energy profile of an important residue, E153 (Supplementary Table 2). Indeed, based on the proposed mechanism and crystal structure of pMT1[13], E153 is the key catalytic residue that extracts the proton from psi-ionone (Supplementary Fig. 5a). The free energy contribution of E153 appears improved in pMT3 compared to pMT1 to favour binding to cis-isomers, mainly (1R, 5S)-cis-α-irone (1R5S), in accordance with results shown in Fig. 2b. Chiral GC analysis showed that 1R5S was predominantly produced by reaction with pMT3 (Supplementary Fig. 5b). Free energy calculations also confirmed the favourable contributions of the two mutated residues in pMT3, namely Y200F and S182E (Supplementary Table 2). For all MD simulations with pMT3, Y200F and S182E formed an energetically and structurally favourable environment to interact with α-irone. This environment was structurally maintained by L273, which was proposed for the next round of mutation (Supplementary Fig. 6a). Mutating L273 to lysine or

valine further improved enzymatic selectivity; L273V (pMT4) improved cis-α-irone content to ~68.0% by reducing trans-α-irone production (Fig. 2b and Supplementary Fig. 6b). These results converge with the computational predictions for pMT4, which showed an increase of the free energy contribution of E153 toward trans-isomers compared to pMT1 and pMT3 mutants (Supplementary Table 2). When combined with L180A (pMT5), the enzyme expression was improved, and the cis-α-irone content was further increased to ~82.2% (Supplementary Fig. 3a and Fig. 2b).

Moreover, since psi-ionone (556.3 $Å^2$) is smaller than pMT1's natural substrate (teleocidin A1, 1129.2 $Å^2$), we mutated A202 to bulkier amino acids (leucine, valine or phenylalanine) based on pMT5 to shrink the binding pocket, in order to increase the affinity of the enzyme for the substrate. Among the new mutants, A202L (pMT6) increased the cis-α-irone concentration (Fig. 2b). At the same time, we observed an additional peak corresponding to β-irone (~30% of the irone mixture, Supplementary Fig. 7). This led us to examine any mutation(s) that would shift the enzymatic selectivity from β-irone to cis- α-irone.

Due to the catalytic role of E153, the mutation of surrounding residues is expected to influence enzyme selectivity. Eighteen amino acid residues were identified in the close vicinity of E153. Pooled experiments were repeated using pMT6 as a template and a E153 mutation was included as baseline activity (Supplementary Fig. 2c). Among them, mutations of Y65 showed the lowest β-to-cis-α-irone ratio, and C156 mutations produced the highest amount of cis-α-irone (Supplementary Fig. 2c). Site-saturation mutagenesis of Y65 led to the discovery of Y65F (pMT7), which increased product selectivity towards cis-α-irone, consequently increasing cis-α-irone content to ~93.1% (Fig. 2b). The cis-α-irone production and TTN of pMT7 were ~2434.5-fold and ~968.4-fold higher compared to pMT1, respectively (Fig. 2b and Supplementary Fig. 3b). The cis-to-trans-α-irone ratio increased by >1000-fold (Fig. 2b). Structurally, this mutation removed the hydrogen bond between Y65 and E153, which enabled the catalytic residue to be closer to the C4 hydrogen of psi-ionone (Supplementary Fig. 6a). We also tested site-saturation mutagenesis on C156 residue based on pMT6, and C156P (pMT8) further improved the TTN of the enzyme (Supplementary Fig. 3b). However, it did not improve product selectivity and exacerbated protein expression (Fig. 2b and Supplementary Fig. 3a). Mutating C156 to proline based on pMT7 resulted in lower methyltransferase activity (Supplementary Fig. 8a). Instead, C156A mutation (pMT9) based on pMT7 increased both α- and β-irone production (Supplementary Fig. 8b). Since pMT7 was active and selective towards cis-α-irone, we further characterized pMT7 in vitro.

## In vitro biotransformation of psi-ionone to cis-α-irone

The steady-state kinetic parameters of pMT7 were determined (Table 1). The $k_{cat}$ and $K_m$ values for catalysing psi-ionone were 0.0022 $h^{-1}$ and 24.9 µM, while those for SAM were 0.0026 $h^{-1}$ and 53.6 µM. While determining the steady-state kinetics of pMT7, we noticed that the reaction with purified pMT7 stopped at 6 h before significant conversion had taken place (Supplementary Fig. 9a). Moreover, while using cell extracts containing pMT7, the reaction plateaued after a 1-day incubation (Supplementary Fig. 9b). Increasing the amount of lysates or prolonging the incubation time did not increase irone production (Supplementary Fig. 9c). Since pMT7 is a hexamer in solution[13], we examined its oligomeric state before and after the reaction. As shown by the native PAGE gel, pMT7 remained intact as a hexamer after incubating or reacting at 28 °C overnight (Supplementary Fig. 9d). Activity was slightly reduced when the enzyme was pre-incubated at 28 °C overnight compared to at 4 °C overnight (Supplementary Fig. 9e), suggesting that pMT7 was stable under the reaction temperature. We then tested if the enzyme was subject to feedback inhibition as the cofactor product SAH is a potent inhibitor of methyltransferases[18]. Supplementing 2 µM of SAH reduced pMT7 activity and 20 µM of SAH nearly abolished the

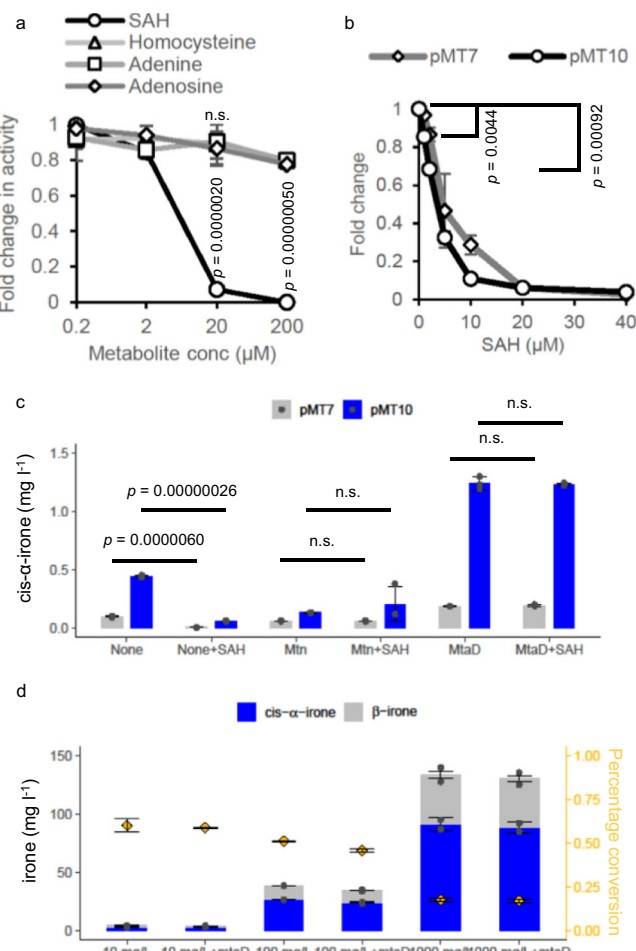

**Fig. 3 | Kinetic characterization of pMT7 and pMT10 enzymes. a** Fold change in pMT7 activity when SAH (0-200 µM), homocysteine (0-200 µM), adenine (0-200 µM), or adenosine (0-200 µM) was supplemented into purified pMT7 reaction. It was calculated by dividing the pMT7 activity where no additive was added. **b** Fold change in pMT7 and pMT10 activities when SAH (0-40 µM) was supplemented into the purified enzyme reactions. It was calculated by dividing the pMT7 and pMT10 activity where no additive was added. **c** Effect of introducing auxiliary enzymes, Mtn and MtaD, to hydrolyse SAH on cis-α-irone production by purifying pMT7 and pMT10 with or without initial 20 µM SAH. pMT10 increased production of cis-α-irone by ~4.5 and ~6.5-fold without and with MtaD enzyme, respectively compared to pMT7 reaction alone. **a** to **c**, The average and s.d. of three biologically independent experiments are shown. Significance (*p*-value) was evaluated by a two-sided Student's *t*-test, n.s. represents *p* > 0.05. **d** In vitro biotransformation of psi-ionone to cis-α-irone by using cell lysates containing overexpressed pMT10. The bar chart represents the mean value of irone concentration. The orange diamonds represent the percentage conversion calculated based on the starting substrate concentration. The average and the s.d. of two biologically independent experiments are shown in the graph. The abbreviations are as follows. Mtn, *S*-adenosylhomocysteine nucleosidase. MtaD, *S*-adenosylhomocysteine deaminase. SAH, *S*-adenosylhomocysteine. Source data are provided as a Source Data file.

methyltransferase activity (Fig. 3a, b). IC₅₀ of SAH for pMT7 was 4.9 µM (Table 1). We also tested supplementing with SAH analogues (homocysteine, adenine, adenosine) and an excess of α-irone, but doing so did not significantly alter the activity of methyltransferase (Fig. 3a and Supplementary Fig. 9f).

To reduce SAH inhibition, we mutated the residues (D135 and T91) that form a hydrogen bond with SAH to weaken the binding between SAH and the enzyme (Supplementary Fig. 10a). Mutating D135 to alanine, glutamate, glycine and proline nearly abolished the enzymatic activity (Supplementary Fig. 10b). Interestingly, mutating T91 to

proline marginally increased activity compared to pMT7 (Supplementary Fig. 10c). We subsequently combined T91P with C156A and created pMT10, which increased cis-α-irone production by ~4.3-fold compared to pMT7 (Supplementary Fig. 10d). Effectively, pMT10 increased cis-α-irone production by >10,000-fold as compared to pMT1 (all the mutations are summarized in Supplementary Fig. 11). Steady-state kinetics analysis showed that k_{cat} of pMT10 (0.0199–0.0258 h⁻¹) was ~10-fold higher than pMT7. Unexpectedly, K_m of SAM for pMT10 (31.4 µM) decreased compared to pMT7. This led to increased sensitivity of pMT10 towards SAH, as shown by a decrease in IC₅₀ of SAH for pMT10 to 3.1 µM (Table 1).

An alternative strategy to reduce SAH inhibition is to introduce auxiliary enzymes that degrade SAH[18] (Supplementary Fig. 12a). As shown in Fig. 3c, supplementary *S*-adenosylhomocysteine nucleosidase (Mtn) and *S*-adenosylhomocysteine deaminase (MtaD) restored pMT7 and pMT10 activity when 20 µM SAH was introduced in the purified enzymatic reaction. However, supplementing Mtn did not enhance the methyltransferase activity, probably due to the low purity of the purified Mtn (Supplementary Fig. 12b). Notably, cis-α-irone production by pMT10 with MtaD improved by ~2.5-fold compared to pMT10 reaction alone (Fig. 3c). MtaD is a deaminase that modifies the adenine moiety[19], suggesting that the amine group in adenine may play an important role in enzyme binding. This observation somewhat agrees with the effect of mutating D135, which abolishes the hydrogen bond interaction between pMT7 and the amine group of the adenine moiety.

With the positive effect of MtaD, we explored in vitro biotransformation by using cell lysate containing overexpressed pMT10 and MtaD. To challenge the enzyme, we supplemented 10 mg l⁻¹ (0.05 mM), 100 mg l⁻¹ (0.5 mM) or 1000 mg l⁻¹ (5 mM) of psi-ionone into the reaction, resulting in the production of 3.2 mg l⁻¹ (-0.015 mM), 26.9 mg l⁻¹ (-0.13 mM) and 91.1 mg l⁻¹ (-0.44 mM) of cis-α-irone, respectively after incubating the reaction at 28 °C for 3 days (Fig. 3d). We observed that the higher the initial concentration of psi-ionone, the lower the percentage of conversion. This could be due to the poor solubility of psi-ionone in an aqueous system. Supplementing MtaD did not further improve cis-α-irone production, suggesting the endogenous Mtn present in *E. coli* extracts was sufficient to alleviate the inhibition of SAH on methyltransferase activity.

## In vivo biotransformation of glucose to cis-α-irone

The advantage of in vivo biotransformation is that microbial cells can regenerate expensive co-factors, hence rendering the scale-up bioprocess more cost-effective. Moreover, microbial cells possess sophisticated mechanisms to regulate SAH concentration[20] and may alleviate the inhibition of SAH to pMT enzyme. Thus, pMT was incorporated into our psi-ionone-producing microbe[10,11], to synthesize cis-α-irone from cheap and simple carbon sources (e.g. glucose/glycerol, Fig. 4a, b and Supplementary Fig. 13). To achieve high productivity, we first optimized psi-ionone production by multidimensional heuristic process (MHP)[21], and identified strain 2O31 to be the highest producer of psi-ionone (Supplementary Fig. 13b). Strain 2O31 was used as the base strain to probe cis-α-irone production in vivo (Supplementary Table 3). Moreover, to improve plasmid stability, an auxotrophic strain was used with chemically defined media[1]. We then transformed the plasmid carrying pMT1, pMT7 or pMT10 into strain 2O31 and produced <0.01, ~1 and ~1.5 mg l⁻¹ cis-α-irone, respectively in 2 days (Supplementary Fig. 13d). To further optimize irone production with pMT10, three different culture conditions were tested: 1) no methionine or dodecane added (control); 2) 10 mM methionine added; and 3) 10 mM methionine and 20% dodecane added. In the control reaction, approximately 2.4 mg l⁻¹ or 0.42 mg l⁻¹ OD⁻¹ cis-α-irone was produced after a 3-day incubation (2O31-pMT10, Fig. 4c and e). Adding 10 mM methionine to increase SAM pool did not increase irone titre and reduced the specific irone titre (2O31-pMT10, Fig. 4c and e). Lastly,

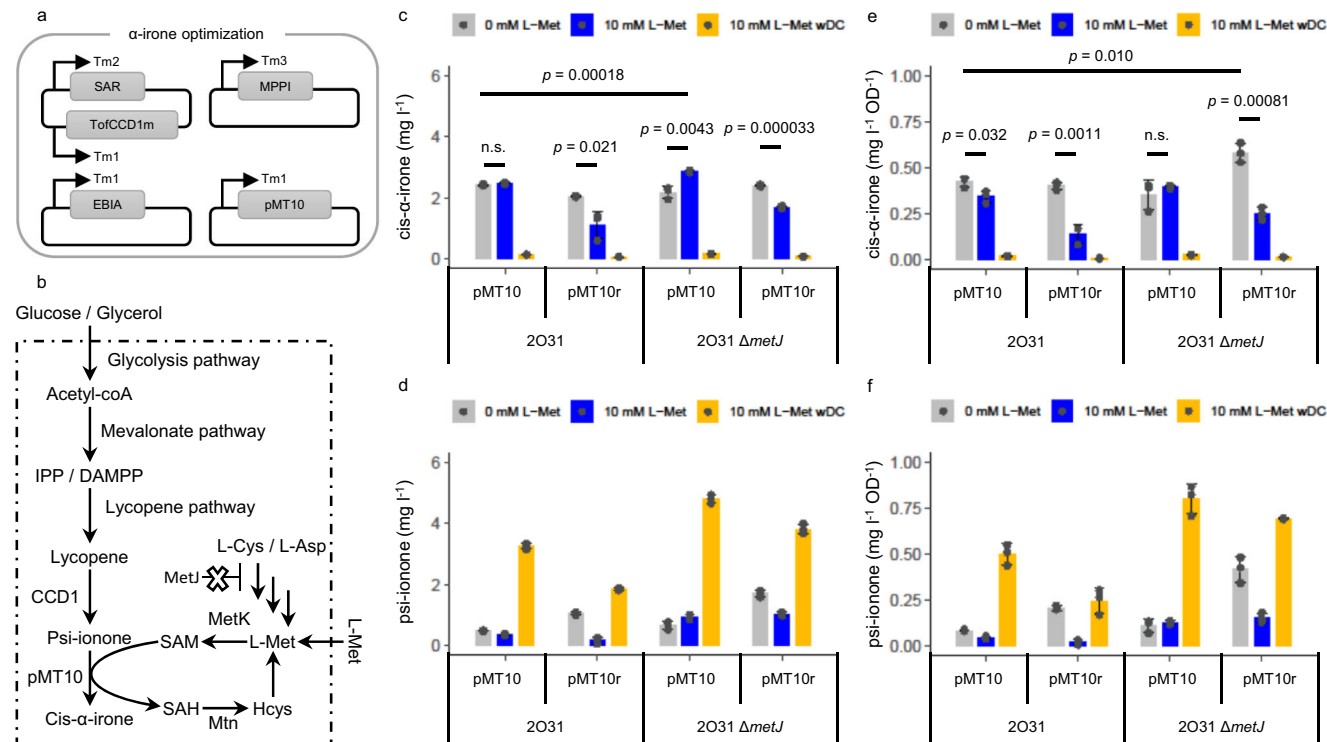

**Fig. 4 | Optimizing in vivo production of cis-α-irone from a simple carbon source. a** Schematic representation of the plasmids used to produce cis-α-irone (Supplementary Table 3). Four plasmids were used. The first plasmid carried the SAR module, which contained the upper mevalonate enzymes: HMG-CoA synthase (HmgS), acetoacetyl-CoA thiolase (AtoB) and truncated HMG-CoA reductase (tHmgR), and engineered OfCCD1 fused with thioredoxin (TofCCD1m)[10]. The second plasmid carried the MPPI module, which contained the lower mevalonate enzymes: mevalonate kinase (MevK), phosphomevalonate kinase (PMK), mevalonate pyrophosphate decarboxylase (PMD) and isopentenyl pyrophosphate (IPP) isomerase (Idi). The third plasmid carried the lycopene synthesis EBIA module: geranylgeranyl pyrophosphate (GGPP) synthase (CrtE), phytoene synthase (CrtB), phytoene desaturase (CrtI) and farnesyl pyrophosphate (FPP) synthase (IspA). The fourth plasmid carried the pMT10 enzyme or pMT10 with the SAM cycle enzymes: S-adenosylmethionine synthase (MetK) and S-adenosylhomocysteine nucleosidase

(Mtn).Tm1, Tm2, Tm3 are the mutated T7 promoters with different strengths (Tm1 > Tm2 > Tm3)[21]. **b** Schematic representation of the pathway to produce cis-α-irone from glucose or glycerol. SAM cycle enzymes (MetK and Mtn) and simplified regulation by metJ are shown. Dotted line represents the cell membrane. The bar charts represent the average amount of cis-α-irone produced in (**c**) the average amount of leftover psi-ionone in (**d**) the average specific titre of cis-α-irone produced in (**e**) the average specific titre of leftover psi-ionone in (**f**). **c**–**f** the average and the s.d. of three biologically independent experiments are shown. For **c** and **e**, significance (p-value) was evaluated by a two-sided Student's t-test, n.s. represents $p > 0.05$. Refer to Supplementary Table 3 for strain-description. Three conditions were tested for in vivo α-irone production. Zero millimolar of L-Met: no methionine or dodecane was added. 10 mM L-Met: 10 mM methionine but no dodecane was added. 10 mM L-Met wDC: 10 mM methionine with 20 % by volume dodecane were added. Source data are provided as a Source Data file.

when both methionine and dodecane were added, psi-ionone accumulated in the organic layer and only ~0.15 mg l⁻¹ or ~0.023 mg l⁻¹ OD⁻¹ cis-α-irone was produced (2O31-pMT10, Fig. 4c–e). This observation suggested that the secretion of psi-ionone was faster than methylation and was facilitated by the dodecane layer. The limited methylation efficiency might be due to insufficient cofactor or SAH inhibition. Hence, we co-expressed MetK and Mtn to convert methionine to SAM and hydrolyse SAH (2O31-pMT10r) and tested the three conditions again. Unexpectedly, cis-α-irone concentration decreased under all three conditions (2O31-pMT10r, Fig. 4c, e). In fact, the total amount of psi-ionone and cis-α-irone decreased when 10 mM methionine was added (Fig. 4c, d). As SAM biosynthesis is regulated by *metJ* in *E. coli*[22,23], we then examined the aforementioned six reactions with *metJ* deleted strain (2O31, ΔmetJ). An increase in cis-α-irone titre to ~2.86 mg l⁻¹ was produced by the 2O31 ΔmetJ-pMT10 strain with 10 mM methionine supplementation (Fig. 4c). An increase in specific cis-α-irone titre to ~0.58 mg l⁻¹ OD⁻¹ was produced by the 2O31 ΔmetJ-pMT10r strain without methionine supplementation (Fig. 4d). We noted that psi-ionone concentration was generally higher in the 2O31 ΔmetJ strain compared to the 2O31 strain, suggesting that irone concentration could be potentially increased (Fig. 4d, f).

Lastly, a single-phase fed-batch process was tested in a 5 l bioreactor. Even though the 2O31 ΔmetJ strain produced a higher titre of

cis-α-irone, its growth seemed to be impeded during fermentation. Thus, the 2O31-pMT10 strain was tested. The optimized bioprocess for α-ionone was applied[11], where the feeding rate of 500 g l⁻¹ glucose was set at 20 ml h⁻¹ and the glucose concentration was kept between 0−5 g l⁻¹ in the bioreactor. Two different dissolved oxygen (DO) levels were tested: 2% and 10%, to minimize product loss due to a high aeration rate. Under these DO settings, evaporation of irone was kept at a minimum (<0.05%). As shown in Fig. 5, cis-α-irone was produced faster at 10% DO than 2% DO on the first day post-induction. Subsequently, cis-α-irone production accelerated at 2% DO than 10% DO. At 113 h, when glucose feeding was stopped, ~68.0 mg l⁻¹ cis-α-irone and ~28.0 mg l⁻¹ β-irone were produced at 2% DO, and ~74.2 mg l⁻¹ cis-α-irone and ~31.3 mg l⁻¹ β-irone were produced at 10% DO. By continuing to incubate the cultures for another 24 h, a final titre of ~76.5 mg l⁻¹ cis-α-irone and 32.5 mg l⁻¹ β-irone were produced at 2% DO and ~86.0 mg l⁻¹ cis-α-irone and 36.6 mg l⁻¹ β-irone were produced at 10% DO. Simultaneously, a decrease in psi-ionone concentration was observed during the last 24 h, indicating that psi-ionone was converted to irones. Future work to enhance the selectivity of pMT10 from β-irone to cis-α-irone could further increase cis-α-irone titre.

Taken together, eight mutations were introduced to pMT1 and improved both activity and product-selectivity by >10,000-fold and

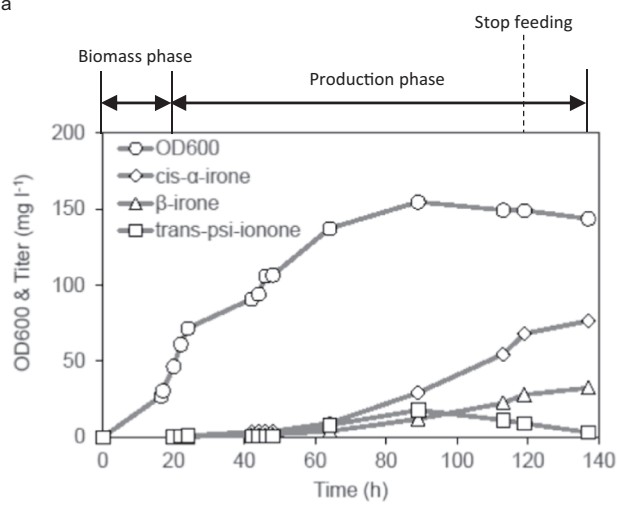

**Fig. 5 | Fed-batch process of cis-α-irone.** Time-course profiles of $OD_{600}$, cis-α-irone, β-irone and psi-ionone titres when **a** DO was controlled at 2%. **b** when DO was controlled at 10% during the production phase. For both bioprocesses, the production was induced when $OD_{600}$ reached 30–40 and glucose feeding was stopped at 113 h. Source data are provided as (**a**).

>1000-fold, respectively (Supplementary Fig. 11 and Supplementary Table 4). Wild-type pMT1 converted psi-ionone to predominantly trans-α-irone, which is odourless. The improved mutant pMT7 converted psi-ionone to >90% cis-α-irone, which has the finest, iris-like notes[8]. By further modulating the SAH binding pocket residues, we obtained pMT10 with a 10-fold improvement in catalytic efficiency (Table 1) and produced ~91.1 mg l$^{-1}$ cis-α-irone from psi-ionone. By incorporating pMT10 into metabolically engineered *E. coli*, ~86.0 mg l$^{-1}$ cis-α-irone was produced from glucose in a fed-batch process. As compared to natural extraction, our current bioprocess is 3800–18,000 –fold higher in yield for the same land size and duration (Supplementary Table 5), paving a way towards sustainable bioproduction of the premium perfume molecule.

## Methods
### Strains, plasmids, and chemicals
*E. coli* Bl21-Gold DE3 strain (Stratagene) was used in this study. CRISRP-cas9 mediated gene deletion was carried out to modify the genome of *E. coli* BL21 strain[24]. pET11a (Novagen) was used to construct the methyltransferase mutants. For in vivo psi-ionone production, the plasmids used contained modified p15A origins of replication (Supplementary table 3)[10]. The genes: *pMT1* or *TleD* from *Streptomyces blastmyceticus*, *SaMT* from *Streptomyces albireticuli*, *ScMT* from *Streptomyces clavuligerus, Mt.CMAS* from *Mycobacterium tuberculosis, Sl.GdpMT* from *Streptomyces lasalocidi*, *Ss.SpnF* from *Saccharopolyspora spinosa*, and *MtaD* from *Thermotoga maritima* were codon optimized and synthesized by Integrated DNA Technologies, Inc. *Mtn* and *MetK* were amplified from the *E. coli* genome. The sequences and associated accession numbers are provided in Supplementary data 3. Unless otherwise noted, all chemicals and reagents were obtained from Sigma-Aldrich.

### Cloning and site-directed mutagenesis
Mutations were carried out with a modified QuikChange™ protocol[10]. Briefly, overlapping primers, which carried the desired mutation, were designed to amplify the plasmid carrying the pMT gene. The plasmid was amplified with polymerase chain reaction (PCR) with high-fidelity iproof polymerase (Bio-Rad) in 10 µl reaction. Subsequently, 0.5 unit of DpnI enzyme (New England Biolabs) was added directly to the PCR reaction mixture to remove the template plasmids at 37 °C for 3 h. Lastly, 1 µl of the mixture was transformed into 20 µl of Stellar competent cells (Clonetech,

Supplementary Table 3) by heat-shock transformation. The mutation was verified by sequencing.

For the pooled colony screening, the cloning steps were the same as the directed mutagenesis except that a degenerative primer with NNK or NNN was used to amplify the plasmid. After transformation, all the colonies were combined with 1 ml phosphate buffer saline solution (PBS) and subjected to plasmid extraction. The purified plasmid was then transformed into BL21 cells and plated on agar with ZYM auto-induction media and incubated at 20 °C for 3 days[25]. The colonies formed were collected in 1 ml PBS, and OD was measured. 12.5 × 250 µl*OD cells were taken for subsequent reaction with psi-ionone to test methyltransferase activity. All the primers used in the study are provided in Supplementary data 3.

### Protein purification
His-tag protein purification was performed according to manufacturer's instruction (Cube Biotech). Each enzyme was overexpressed in 200 ml of ZYM autoinduction media with 5 mM lactose at 20 °C for 36 h. Following that cells were pelleted and resuspended with 10 ml of His-binding buffer (50 mM Tris, pH 8, 0.5 M NaCl and 20 mM imidazole) supplemented with 1 mg ml$^{-1}$ lysozyme and 3 U ml$^{-1}$ of benzonase nuclease (Merck) and treated for 1 h at 20 °C, 300 rpm. The cells were further lysed by 2 cycles of freeze-thawing with freezing at −80 °C for 4 h and thawing at room temperature for 45 mins. The supernatant (soluble fraction) was obtained by centrifuging at 15,000 g for 30 mins at 4 °C. The collected supernatant was then mixed with 1 ml of equilibrated Ni-NTA beads (Cube Biotech) overnight at 4 °C with continuous mixing. The next day, the beads were washed 3 times with 5 ml of His-binding buffer to remove any unbound proteins before eluting bound enzyme 5 times with 0.5 ml of His-elution buffer (50 mM Tris, pH8, 0.5 M NaCl, 0.5 M imidazole). The eluted protein was concentrated using spin-column with a molecular weight cut-off of 10 kDa (Sartorius). Protein quantification and purity were determined through the Micro BCA assay (Pierce) and running sodium dodecyl sulphate–polyacrylamide gel electrophoresis (SDS-PAGE), respectively.

### Purified enzyme reaction
Following enzyme purification, a 100 µl enzymatic reaction was setup with a final concentration of 0.5 mg ml$^{-1}$ of enzyme, 40 mg l$^{-1}$ psi-ionone, 0.2 mM SAM, 100 mM Tris (pH 7), 10 mM MgCl$_2$ and 15 mM NaCl. The reaction was done in 0.2 ml PCR strips and incubated at 28 °C, 1200 rpm overnight. At the same time, a reaction without

enzyme (with 40% glycerol only) was included as a negative control. The next day, cis-α-irone was extracted with 100 µl of ethyl acetate and 50 µl organic layer for gas chromatography−mass spectrometry (GCMS) analysis. For enzyme characterization, similar enzymatic reactions were performed, with varying concentrations of psi-ionone (2–40 mg l$^{-1}$) and SAM (10–100 µM), and the reactions were stopped after 2 h incubation at 28 °C to ensure the enzymatic conversion is still within the linear range. For TTN determination, a similar 1 ml enzymatic reaction was set up in solid phase microextraction (SPME) vials with 0.1 mg ml$^{-1}$ MtaD enzyme. To avoid saturating the GC signal, the pMT enzyme concentration was varied as follows: 1) 3.5 mg ml$^{-1}$ pMT1; 2) 1 mg ml$^{-1}$ pMT2 and pMT3; 3) 0.5 mg ml$^{-1}$ pMT4 and pMT5; and 4) 0.05 mg ml$^{-1}$ pMT6, pMT7, and pMT8. The reaction was incubated at 28 °C for 72 h to ensure that the reactions had been completed.

## Cell extracts preparation and reaction
Each pMT mutant was transformed into BL21 and directly grown in autoinduction media with 5 mM lactose at 20 °C for 36 h. Then, cells were pelleted and concentrated 10 times by adding 1 ml of PBS. The OD600 was measured and 40 × 600 µl*OD cells were transferred into a new 1.5 ml tube. The cells were spun down and resuspended in 600 µl of lysis buffer (10 mM Tris (pH7), 2.5 mM MgCl$_2$, 0.5 mM CaCl$_2$, 150 mM NaCl, 0.5% glycerol, 1 mg ml$^{-1}$ lysozyme and 20 U ml$^{-1}$ of DNaseI). The cells were incubated for 2 h at 20 °C with shaking at 300 rpm. Following that, 2 cycles of freeze-thaw was done to completely lyse the cells. At the end of the 2$^{nd}$ thawing, the cell lysate was used to setup a 1 ml reaction in 2 ml GC vials. The reaction consisted of 50 mM Tris (pH7), 10 mM MgCl$_2$, 60 mM NaCl, 0.2 mM SAM and 10 mg l$^{-1}$ of psi-ionone and caryophyllene each. To this, 500 µl of cell lysate was added while the remaining 100 µl was used for SDS-PAGE and western blot to analyse the total and soluble protein expression. The reaction was incubated for 2 days at 28 °C, 300 rpm prior to extraction of cis-α-irone with 500 µl hexane. From this, 20 µl of the organic layer was taken into 180 µl hexane for GCMS analysis.

## Western blot analysis
Standard SDS-PAGE protocol was performed before transferring proteins onto nitrocellulose membrane using the iBolt2 dry transfer system (Thermofisher). The membrane was then blocked with 20 ml of 5% milk in tris-buffered saline with 0.1% Tween® 20 detergent (TBST) buffer for 1 h at room temperature. Following that, it was probed with 10 ml HRP Anti-6X His tag® antibody [GT359] (ab184607, Abcam) diluted 2000 times in 1% milk in TBST buffer overnight at 4 °C. The membrane was washed 3 times with 20 ml of TBST buffer before adding in the substrate for chemiluminescence detection (Millipore) and imaging using the ChemiDoc system (Bio-Rad). The chemiluminescence signal was quantified using Image Lab (Bio-Rad). Purified pMT at known concentrations (2.8–90 ng µl$^{-1}$) was used to validate the antibody.

## Biotransformation of glucose/glycerol into cis-α-irone
The four plasmids were transformed into *E. coli* BL21 (DE3), ΔaroA, ΔaroB, ΔaroC, ΔserC or *E. coli* BL21 (DE3), ΔaroA, ΔaroB, ΔaroC, ΔserC, ΔmetJ strains and plated on agar plates containing LB media (10 g tryptone, 5 g yeast extract, and 10 g NaCl) supplemented with the appropriate antibiotics (100 mg l$^{-1}$ ampicillin, 34 mg l$^{-1}$ chloramphenicol, 50 mg l$^{-1}$ kanamycin and 100 mg l$^{-1}$ spectinomycin). One colony was randomly picked and inoculated into LB media with antibiotics. 1% of overnight culture was inoculated into 1 ml fresh auto-inducing chemically defined media. The chemical media contains a carbon source solution (0.5 g l$^{-1}$ glucose, 10 g l$^{-1}$ glycerol), inducer (30 mM lactose), base media (2 g l$^{-1}$ ammonium sulphate, 4.2 g l$^{-1}$ KH$_2$PO$_4$, 11.24 g l$^{-1}$ K$_2$HPO$_4$, 1.7 g l$^{-1}$ citric acid, 0.5 g l$^{-1}$ MgSO$_4$), and 10 ml l$^{-1}$ trace element solution. The trace element solution (100×) contained 0.25 g l$^{-1}$ CoCl$_2$·6H$_2$O, 1.5 g l$^{-1}$ MnSO$_4$·4H$_2$O, 0.15 g l$^{-1}$

CuSO$_4$·2H$_2$O, 0.3 g l$^{-1}$ H$_3$BO$_3$, 0.25 g l$^{-1}$ Na$_2$MoO$_4$·2H$_2$O, 0.8 g l$^{-1}$ Zn(CH$_3$COO)$_2$, 5 g l$^{-1}$ Fe(III) citrate, and 0.84 g l$^{-1}$ ethylenediaminetetra-acetic acid (EDTA) at pH 8.0[21]. The culture was grown at 28 °C for 3 days and the products were extracted with 1 ml hexane for GCMS analysis. An optimized fed-batch process for α-ionone was used for the bioreactor test[11]. Briefly, overnight culture was inoculated into 2 l chemical media comprising the base media and 5 g l$^{-1}$ glucose, at 37 °C and 30% DO. pH was controlled at 7 with base solution (14% ammonia and 0.5 M NaOH). Feed media (500 g l$^{-1}$ glucose and 5 g l$^{-1}$ MgSO$_4$) was added into bioreactor once OD$_{600}$ reached 5, at a rate of 14.3-51.5 ml h$^{-1}$ glucose for 4–5 h until OD$_{600}$ reached ~30–40. Subsequently, 0.1 mM IPTG was added to induce the production, feeding rate was kept at a constant rate of 20 ml h$^{-1}$ glucose, and the temperature was reduced to 30 °C. DO was adjusted to 2% or 10%. Feeding was stopped at 113 h when the 2 l feeding media was depleted, but the culture was further incubated for 24 h. The exhaust was connected to 25 ml of sunflower oil to capture the evaporated products.

## Gas chromatography mass spectrum analysis
For the mutant pMT activity screening, the reactants and products were analysed by head space solid phase microextraction (HS-SPME) coupled with an Agilent 5977B gas chromatography (GC) system equipped with Agilent DB-5ms column (30 m × 250 µm × 0.25 µm) and mass spectrometry (MS) with a high-efficiency source. The reactions were carried out in SPME vials that were then incubated at 60 °C for 20 min to release the volatile compounds to the headspace. The compounds were then extracted by the absorbent fibre (50/30 µm divinylbenzol/carboxen/polydimethyl-SPME fibre, SUPELCO) for 20 min, desorbed at the GC inlet at 250 °C for 1 min and injected into GC with a split ratio of 200:1. The GC oven temperature was increased from 50 °C to 140 °C at a rate of 10 °C min$^{-1}$ and held at 140 °C for 10 min. Subsequently, the temperature was increased to 320 °C at a rate of 60 °C min$^{-1}$ and held at 320 °C for 2 mins. The concentration of psi-ionone and irone were calculated against a standard curve prepared with synthetic standards.

To determine the purified enzyme kinetics and in vivo production, the reaction mixture was extracted with 0.5× or 1× volume of ethyl acetate and the organic phase was diluted appropriately before being subjected to GCMS analysis. The samples were analysed on an Agilent Intuvo 9000 GC system equipped with Agilent DB-WAX Ultra Inert Intuvo GC column (30 m × 250 µm × 0.25 µm) and Agilent 5977B mass spectrometry with a high-efficiency source. 1 µl organic phase was injected at the split ratio of 10:1 at 250 °C. The oven temperature was held at 50 °C for 1 min and increased to 200 °C at a rate of 40 °C min$^{-1}$ and held at 200 °C for 3 min. Subsequently, the temperature increased to 230 °C at a rate of 40 °C min$^{-1}$ and held at 230 °C for 5 mins. The concentration of psi-ionone and irone was calculated against a standard curve prepared with synthetic standards.

## Computational studies
The three-dimensional (3D) structure of pMT1 in complex with the cofactor SAH with the substrate teleocidin A1 was retrieved from the PDB database (PDB id: 5GM2). One dimeric unit was used as the template for modelling pMT1 in complex with the four α-irone isomers (1R5R and 1S5S for the trans- α-irone and 1S5R and 1R5S for the cis- α-irone). The α-irones were first built and gradually relaxed using the Avogadro software version 1.1.1[26]. They were then manually docked into the pMT1 active site using the bound substrate teleocidin A1 as a template from crystallographic structure. pMT3 and pMT4 were modelled using Modeller 9.19[27] and 3D models with the lowest DOPE score were kept for further analyses. The corresponding complexes of pMT3 or pMT4 with the four α-irones were built by superposing pMT1 complexes.

MD simulations were performed using the AMBER ff14S force-field[28] for enzymes and GAFF[29] for the ligands using pmemd.CUDA in

AMBER18 software[30]. The partial charges for the ligand were computed using the AM1-BCC method from the antechamber package[31]. The system was protonated using propka webserver to set the experimental pH of 7. The MD simulation parameters[32] were as follows: In all stages, the sulphur of SAH was kept close to the key carbon irone C4 (Supplementary Fig. 5) isomer using NMR distance restraint algorithm, with the following parameters on the distance in Å ($r1 = 0$, $r2 = 0$, $r3 = 4$, $r4 = 5$) and following parameters on the force constant in kcal mol$^{-1}$ Å$^{-2}$ ($rk2 = 0$, $rk3 = 25.0$). The MD simulations were carried out for a total of 5 ns for all complexes. MM/PBSA calculations were performed using MMPBSA.py software and default parameters for Poisson–Boltzmann as described by Miller et al.[33] Standard parameters were used such as $igb = 5$ for the generalized Born model and $saltcon = 0.1$ for the salt concentration. The pdb files for the protein structures are provided as Supplementary data 1 and 2.

## Reporting summary

Further information on research design is available in the Nature Portfolio Reporting Summary linked to this article.

## Data availability

All data supporting the findings of this study are available in the article, its supplementary information, and supplementary datas, or upon request from the corresponding authors. The source data for all figures reported in the article and its supplementary information are provided in the Source Data file. The protein structure data generated in this article is available in World wide Protein Data Bank under PDB ID 5GM1 (https://www.wwpdb.org/pdb?id=pdb_00005gm1), 5GM2 (https://www.wwpdb.org/pdb?id=pdb_00005gm2), 1KPG (https://www.wwpdb.org/pdb?id=pdb_00001KPG), 4F86 (https://www.wwpdb.org/pdb?id=pdb_00004F86), 4PNE (https://www.wwpdb.org/pdb?id=pdb_00004PNE) and Supplementary data 1. The initial and final configurations for molecular dynamics calculations are provided in Supplementary data 2. The sequence data, primers and gRNA for CRISPR-cas9 guided knockout used in this article are provided in Supplementary data 3. Source data are provided with this paper.

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

## Acknowledgements

The authors would like to thank Dr. Nicholas David Lindley from the Singapore Institute of Food and Biotechnology Innovation, and Dr. Magali Remaud-Simeon, Dr. Sophie Bozonnet from the Toulouse Biotechnology Institute for their valuable suggestions. We acknowledge Dr Ee Lui Ang, and Dr Hazel Khoo, the Singapore Institute of Food and Biotechnology Innovation for supporting this project. This research is supported by the Agency for Science, Technology, and Research (A*STAR) under IAFPP3 - H20H6a0028, AME Young Individual Research Grants: A1984c0040 (2018) to X.C. and A2084c0064 (2019) to Z.C.

## Author contributions

X.C. conceived the idea of artificial pathway design. X.C., R.T., S.S., C.L. and L.O. performed the experiments. J.E., D.S. and I.A. performed the in silico analysis. X.C., R.T., Z.C., J.E. and I.A. wrote the manuscript. X.C., Z.C., J.E. and I.A. reviewed and revised the manuscript. All authors have read and approved the final manuscript.

## Competing interests

A PCT patent application (PCT/SG2022/050235) has been filed through the Agency for Science, Technology and Research and is pending. The inventors are Congqiang Zhang, Derek John Smith, Isabelle André, Jeremy Esque, Rehka T, Sudha Devi D/O Manbahal Shukal and Xixian Chen. All the results presented in the manuscript have been covered in the patent application.
