## [Peer Review File · Nature Communications]

Total enzymatic synthesis of cis- α -irone from a simple carbon sourceREVIEWER COMMENTS

Reviewer #1 (Remarks to the Author):

I think that the paper under review deserves publication in Nature Communications.

The topic is of interest for the journal readers and the described findings are scientifically sound. I have to state that site-directed mutagenesis is not my main field of interest. Therefore my review focuses more on the enzymatic synthesis of fragrances.

Overall, I deem that the paper is well written and I suggest correcting only few inaccuracies, in order to improve the quality of the work.

More specifically, I listed below some suggestions:

-The authors investigated the biotechnological production of irone isomers by engineered *E. coli* cells. They selected cis- α irone as target compound. This is correct because the two enantiomeric form of this irone isomer are actually the most sought after among synthetic irone. In spite of this fact, a third isomer (gamma-irone) is very relevant from an olfactory point of view (Brenna, G.; Delmonte, M.; Fuganti, C.; Serra, S. Enzyme-mediated preparation of (+)- and (-)-beta-irone and (+)- and (-)-cis-gamma-irone from irone alpha (α). *Helv Chim Acta* 2001, 84, 69-86.). Different studies have underlined the importance of both alpha and gamma isomers in orris oil. In light of these facts, why the authors didn't investigate the possible formation of gamma isomers from psi-ionone? Weren't them present in the reaction mixture? The authors should clarify this point.

-Pag. 9. Lines 237-242. The authors stated: 'To challenge the enzyme, we supplemented 10 mg/L (0.05 mM), 100 mg/L (0.5 mM) or 1000 mg/L (5 mM) of psi-ionone into the reaction, and 6.4 mg/L (~0.03 mM), 54 mg/L (~0.26 mM) and 182 mg/L (~0.88 mM) of cis- α -irone were produced, respectively after incubating the reaction at 28 C for 3 days (Fig. 3d). It was observed that the higher the initial concentration of psi-ionone, the lower the percentage of conversion ... ' This reasoning would be corrected only if psi-ionone was soluble in water. I suppose that no more than 10 mg/L of this compound dissolved in water. Therefore is not possible to correlate the reaction rate with reagent concentration for a heterogeneous mixture.

-The authors highlighted the possibility of producing alpha-irone from renewable carbon sources (glucose or glycerol). Since irone is a niche product with high unit value, the cost of the carbon sources used for the medium preparation is negligible if compared with the value of the product. I suggest removing the above mentioned statements.

Reviewer #2 (Remarks to the Author):

Heterologous synthesis of plant-derived natural products in microbial hosts has been recognized as an attractive method in recent years. However, the unclear biosynthetic pathway and the low activities of plant enzymes in microbial hosts especially the prokaryotes still highly limit their production capacity. In this study, the authors proposed a retrobiosynthetic pathway to synthesize a top-end perfumery molecule Cis- α -irone, for which the core is the improvement of a promiscuous methyltransferase by structure-guided enzyme engineering strategies. By rational analysis and rewiring, they greatly improved the enzyme activity and specificity. This allowed the further transformation of certain substrate to cis- α -irone in vitro. Then the enzyme was expressed in a previously constructed substrate producing *E. coli* strain, and produced cis- α -irone successfully. With some steps of culture optimizations, they finally realized a lab-scale bioreactor production of this valuable compounds with a relatively high titre. It represents an attractive work in this research area, especially the full enzyme evolution process. There are some minor issues need to be addressed.

1. By what criteria the authors selected the two dissolved oxygen levels of 2% and 10% during bioreactor fermentation? First, either 2% or 10% is a relative oxygen levels, the real oxygen concentration will change with aeration and agitation from the referred literature. Second, how did the

authors control the constant 2% and 10% dissolved oxygen levels? Generally, it should be quite difficult to control such kind of DO levels in practice. For example, a very small fluctuation may easily lead a variation DO range of 2%~10%. Third, why did you use so low DO levels as *E. coli* is commonly aerobic culture.

2. Line 266, 'this observation suggested that psi-ionone was secreted to the media first and subsequently taken up by the cells for methylation.' The titre in dodecane seems very low, I just wonder that psi-ionone in the organism dodecane layer is possible from dead cells? Also, is the dodecane harmful to *E. coli* cells? Did the authors test the product content intracellular?

3. I don't quite agree with that glucose or glycerol is renewable carbon source.

Reviewer #3 (Remarks to the Author):

The article by Chen and co-workers describes the application of protein and metabolic engineering for the production of the high-value perfume ingredient alpha-cis-irone in engineered *E. coli*. Because the biosynthetic pathway leading to alpha-cis-irone in plants is not yet known, the authors develop an artificial pathway by identifying a promiscuous C-methyltransferase that can convert psi-ionone to cis and trans irones. By an elegant and comprehensive protein engineering effort, the authors achieve to improve the activity and product specificity of the methyltransferase to achieve efficient bioconversion of psi-ionone. They then employ a psi-ionone producing *E. coli* strain as a base to implement the complete biosynthesis of cis-alpha-irone from sugar. This work presents an excellent and high successful protein engineering effort for the construction of an artificial pathway for cis-alpha irone. However, several conclusions about the role and contribution of specific residues of TleD/pMT1 in its mechanism and the effect of their substitution in catalytic activity and product selectivity need to be further substantiated in order for this work to be published.

1. The authors construct several successful mutants in the process of improving catalytic activity and specificity of TleD/pMT1. They also make several suggestions/conclusions about the functional role and effect of each of the corresponding residues in the interaction with the substrate and product. This elaborate work is an important component of the manuscript and the conclusions drawn are key elements of the work. However, the conclusions made are not supported by detailed biochemical data, except from hypotheses based on the available structural information. The authors only report kinetic analysis of mutant pMT7, but do not isolate or characterize any of the previous 6 variants (the experiments were performed using cell extracts). Moreover, determination of turnover numbers and conclusions drawn from those have been made on the basis of the intensity of Western-blotting bands using "equally loaded" SDS-PAGE. Since the conclusions made are central to the manuscript, it is essential that the authors characterize the kinetic properties of these variants, using in vitro assays with purified proteins.

2. The whole approach, starting with the decision to base the artificial pathway on psi-ionone and then the selection of TleD as the basis of the study is not clearly explained. Particularly the identification of TleD is mentioned as serendipitous by the authors. What other methyltransferase were considered. There are several reported to target isoprenoid precursors. Was any of these enzymes tested?

3. In lines 49-57, the authors report part of their results already in the introduction. The manuscript must be reorganized so that these results become the first paragraph of the Results section.

4. Figure 2 contains panels with limited information (such as panel a), while Fig S2 in the supplementary information contains important detailed information that should be part of the main manuscript.

5. How does the mechanism of cyclization by TleD in teleocidin synthesis compare with the mechanism of converting psi-ionone to irones? Can the authors provide a comparison of the catalytic steps and draw analogies /similarities?

6. Reference 1 is hardly relevant for the argument used. The authors must consider references that clearly describe the field and not only cite their work.

7. Lines 24-25. The authors term this approach integrated retrobiosynthetic pathway design. This may be exaggerated considering that the discovery of pMT1, was, according to the authors, serendipitous.

8. Line 70 and in several other places in the main text and SI. The authors use the term “directed evolution” when describing their approach. In my opinion, the approach is based on rational design and I do not see basic elements of directed evolution being implemented here. The use of saturation mutagenesis is common to many protein engineering approaches.

9. What is the send band in the western blot in figure S4? A fragmentation product? Why is it not present in the lanes with the wild-type enzyme?

10. If the lanes in Fig S2 are all equally loaded, then do the different levels of the different indicate? Is this variation due to protein stability or is the variation in gene expression so large between samples? As discussed above, it is essential to isolate and characterize all these variants using in vitro assays to be able to make meaningful conclusions about the results.

We wish to thank the reviewers for their insightful comments and invaluable suggestions. We have revised the manuscript and have addressed all queries in detail. The point-by-point response to the reviewers' comments are included below.

REVIEWER COMMENTS

Reviewer #1 (Remarks to the Author):

I think that the paper under review deserves publication in Nature Communications. The topic is of interest for the journal readers and the described findings are scientifically sound. I have to state that site-directed mutagenesis is not my main field of interest. Therefore my review focuses more on the enzymatic synthesis of fragrances. Overall, I deem that the paper is well written and I suggest correcting only few inaccuracies, in order to improve the quality of the work. More specifically, I listed below some suggestions:

-The authors investigated the biotechnological production of irone isomers by engineered *E. coli* cells. They selected cis-alpha irone as target compound. This is correct because the two enantiomeric form of this irone isomer are actually the most sought after among synthetic irone. In spite of this fact, a third isomer (gamma-irone) is very relevant from an olfactory point of view (Brenna, G.; Delmonte, M.; Fuganti, C.; Serra, S. Enzyme-mediated preparation of (+)- and (-)-beta-irone and (+)- and (-)-cis-gamma-irone from irone alpha (r). *Helv Chim Acta* 2001, 84, 69-86.). Different studies have underlined the importance of both alpha and gamma isomers in orris oil. In light of these facts, why the authors didn't investigate the possible formation of gamma isomers from psi-ionone? Weren't them present in the reaction mixture? The authors should clarify this point.

We would like to thank the reviewer's suggestion. We agree that gamma-irone is another interesting and valuable target. However, based on our GCMS analysis, gamma-irone is not detected from our reaction mixture. For this manuscript, our main target is cis-alpha-irone. Gamma-irone could be the next target, which requires fine-tuning of the catalytic site of the methyltransferase to convert psi-ionone to gamma-irone.

-Pag. 9. Lines 237-242. The authors stated: 'To challenge the enzyme, we supplemented 10 mg/L (0.05 mM), 100 mg/L (0.5 mM) or 1000 mg/L (5 mM) of psi-ionone into the reaction, and 6.4 mg/L (~0.03 mM), 54 mg/L (~0.26 mM) and 182 mg/L (~0.88 mM) of cis- α -irone were produced, respectively after incubating the reaction at 28 C for 3 days (Fig. 3d). It was observed that the higher the initial concentration of psi-ionone, the lower the percentage of conversion ... ' This reasoning would be corrected only if psi-ionone was soluble in water. I suppose that no more than 10 mg/L of this compound dissolved in water. Therefore is not possible to correlate the reaction rate with reagent concentration for a heterogeneous mixture.

We would like to thank the reviewer's valuable suggestion and we have modified the text accordingly (line 253, the modified text is highlighted in red) to include a discussion that the decrease in reaction conversion is probably due to the poor solubility of psi-ionone in aqueous system.

-The authors highlighted the possibility of producing alpha-irone from renewable carbon sources (glucose or glycerol). Since irone is a niche product with high unit value, the cost of the carbon sources used for the medium preparation is negligible if compared with the value of the product. I suggest removing the above mentioned statements.

We agree with the reviewer that based on the cost of iron, the cost of carbon source may become negligible. We have modified the sentence throughout the manuscript and changed it to simple carbon source instead (the modified text is highlighted in red). This is to differentiate our method from other orris-plant dependent biotransformation method.

Reviewer #2 (Remarks to the Author):

Heterologous synthesis of plant-derived natural products in microbial hosts has been recognized as an attractive method in recent years. However, the unclear biosynthetic pathway and the low activities of plant enzymes in microbial hosts especially the prokaryotes still highly limit their production capacity. In this study, the authors proposed a retrobiosynthetic pathway to synthesize a top-end perfumery molecule Cis- α -iron, for which the core is the improvement of a promiscuous methyltransferase by structure-guided enzyme engineering strategies. By rational analysis and rewiring, they greatly improved the enzyme activity and specificity. This allowed the further transformation of certain substrate to cis- α -iron in vitro. Then the enzyme was expressed in a previously constructed substrate producing *E. coli* strain, and produced cis- α -iron successfully. With some steps of culture optimizations, they finally realized a lab-scale bioreactor production of this valuable compounds with a relatively high titre. It represents an attractive work in this research area, especially the full enzyme evolution process. There are some minor issues need to be addressed.

1. By what criteria the authors selected the two dissolved oxygen levels of 2% and 10% during bioreactor fermentation? First, either 2% or 10% is a relative oxygen levels, the real oxygen concentration will change with aeration and agitation from the referred literature. Second, how did the authors control the constant 2% and 10% dissolved oxygen levels? Generally, it should be quite difficult to control such kind of DO levels in practice. For example, a very small fluctuation may easily lead a variation DO range of 2%~10%. Third, why did you use so low DO levels as *E. coli* is commonly aerobic culture.

We would like to thank the reviewer's valuable comments. We agree that *E. coli* is aerobic culture, thus DO was kept at 30% for cell growth during biomass phase (Figure R1). DO was changed to 2% and 10% during the production phase, namely after IPTG was added. During production phase, glucose feeding was kept constant and limiting, thus the DO fluctuation remained relatively small (Figure R1). We have modified the text at line 298 (highlight in red) that we chose low DO setting to avoid loss of the product, as high aeration or agitation may lead to air stripping of the product.

Figure R1. Dissolved oxygen for fed-batch process of cis- α -iron. Time-course profiles of DO for **a**, DO was controlled at 2%. **b**, when DO was controlled at 10%.

2. Line 266, ‘this observation suggested that psi-ionone was secreted to the media first and subsequently taken up by the cells for methylation.’ The titre in dodecane seems very low, I just wonder that psi-ionone in the organism dodecane layer is possible from dead cells? Also, is the dodecane harmful to *E. coli* cells? Did the authors test the product content intracellularly?

We thank reviewer’s comment, and we agree that it could be due to cell lysis and hence psi-ionone is released to the dodecane layer. However, based on the intracellular metabolite measurement which showed that < 3% psi-ionone is present intracellularly and majority is in dodecane layer (Figure R2a), we believe psi-ionone in dodecane layer is not entirely from cell lysis but secreted by *E. coli*. The low concentration of psi-ionone production was probably because of the enzyme, CCD1, has not been optimized to convert lycopene to psi-ionone. Indeed, CCD1 is a promiscuous enzyme able to cleave lycopene at various double bond position to form 6-Methyl-5-hepten-2-one, citral and psi-ionone¹. This will be our future study to further enhance psi-ionone production and hence irone production. In addition, cell lysis is unlikely due to the presence of dodecane as the final OD for cultures with or without dodecane are comparable (Figure R2b). This agrees with many literatures which authors used dodecane to trap secreted products from *E. coli*^{2,3}. In fact, study has showed that dodecane has an additional benefit as oxygen-vectors to increase the mass transfer of oxygen for aerobic microbial culture⁴.

Figure R2. In vivo production of cis-α-irone from glucose. **a**, bar chart represents the average metabolite concentration intracellularly, in the media, in dodecane layer and total culture (intracellular+media+dodecane). The intracellular metabolite concentration was measured by collecting the cell pellet, washing with phosphate saline buffer three times and extracting with HAE (hexane : acetone : ethanol ratio to be 2:1:1) solvent. The media was collected and extracted with equal volume of hexane. The dodecane layer was diluted 5x in hexane. The media and total metabolite concentration was measured by adding equal volume of hexane to media or entire culture, respectively. **b**, Bar chart represents the average biomass measured at OD₆₀₀ over triplicated reaction. 10 mM L-Met: 10 mM methionine but no dodecane was added. 10 mM L-Met wDC: 10 mM

methionine with 20 % v/v dodecane were added. For both **a** and **b**, the average and the standard deviation of three biologically independent experiments are shown. Significance (p-value) was evaluated by two-sided student's t-test: ns, P>0.05; * p<0.05; ** p<0.01; *** p<0.001; **** p<0.0001.

3. I don't quite agree with that glucose or glycerol is renewable carbon source.

We thank reviewer's comment and we have changed renewable carbon source to simple carbon source throughout the manuscript (the modified text is highlighted in red). This is to differentiate our method from other biotransformation methods that are orris-root dependent.

Reviewer #3 (Remarks to the Author):

The article by Chen and co-workers describes the application of protein and metabolic engineering for the production of the high-value perfume ingredient alpha-cis-irone in engineered E. coli. Because the biosynthetic pathway leading to alpha-cis-irone in plants is not yet known, the authors develop an artificial pathway by identifying a promiscuous C-methyltransferase that can convert psi-ionone to cis and trans irones. By an elegant and comprehensive protein engineering effort, the authors achieve to improve the activity and product specificity of the methyltransferase to achieve efficient bioconversion of psi-ionone. They then employ a psi-ionone producing E. coli strain as a base to implement the complete biosynthesis of cis-alpha-irone from sugar. This work presents an excellent and high successful protein engineering effort for the construction of an artificial pathway for cis-alpha irone. However, several conclusions about the role and contribution of specific residues of TleD/pMT1 in its mechanism and the effect of their substitution in catalytic activity and product selectivity need to be further substantiated in order for this work to be published.

1. The authors construct several successful mutants in the process of improving catalytic activity and specificity of TleD/pMT1. They also make several suggestions/conclusions about the functional role and effect of each of the corresponding residues in the interaction with the substrate and product. This elaborate work is an important component of the manuscript and the conclusions drawn are key elements of the work. However, the conclusions made are not supported by detailed biochemical data, except from hypotheses based on the available structural information. The authors only report kinetic analysis of mutant pMT7, but do not isolate or characterize any of the previous 6 variants (the experiments were performed using cell extracts). Moreover, determination of turnover numbers and conclusions drawn from those have been made on the basis of the intensity of Western-blotting bands using "equally loaded" SDS-PAGE. Since the conclusions made are central to the manuscript, it is essential that the authors characterize the kinetic properties of these variants, using in vitro assays with purified proteins.

We thank the reviewer's comment. The use of cell lysate to quantify TTN is following Prof Francis Arnold's nature publication⁵. According to Reviewer's suggestion, we have purified all the 8 pMT variants and characterized their total turnover number (Figure R3). The result is included in the new supplementary figure S3b. We did not pursue the km and kcat measurement for all the mutants, as the earlier generations of pMT enzymes, namely pMT1-6, have low activity, resulting in analytical difficulties to quantify irone accurately if we are to vary the substrate concentrations. Moreover, we strongly feel that the measurement of kinetic parameters do not generate additional insights for the eventual application of the enzyme. The eventual use of pMT is to be incorporated into metabolically engineered cells to produce irone from simple carbon source. Thus, we would argue strongly that activity

comparison based on equal amount of cells is more meaningful, as biomass and soluble protein expression per cell are also critical factors that would contribute to the final titer of irone. This is also supported by a report which demonstrated that fermentation yield in terms of final achievable cell concentration and biocatalysts expression level are crucial for decreasing the total cost contribution of the biocatalyst to the final product⁶. The results reported in figure 2b in the main text is normalized based on cell number. The cis- α -irone production increase would take into account of the protein expression level and turnover number. It is closer to the eventual use of pMT. Therefore, we stand our results reported in figure 2b in the main text.

Figure R3. Bar chart representing the mean total turnover number (TTN) of each pMT mutants over replicated reactions. TTN was obtained by dividing the concentration of cis- α -irone produced by the concentration of purified pMT enzyme. Fold change in TTN of all the mutants was measured against the TTN of pMT1, and is represented by blue diamond. Significance (p-value) was evaluated by two-sided student's t-test: ns, $P > 0.05$; * $p < 0.05$; ** $p < 0.01$; *** $p < 0.001$; **** $p < 0.0001$.

2. The whole approach, starting with the decision to base the artificial pathway on psi-ionone and then the selection of TleD as the basis of the study is not clearly explained. Particularly the identification of TleD is mentioned as serendipitous by the authors. What other methyltransferase were considered. There are several reported to target isoprenoid precursors. Was any of these enzymes tested?

We have modified the text to include more description of the pathway design (Line 91-95, highlighted in red). Based on our pathway design, we were looking for a bifunctional methyltransferase and cyclase (bMTC) that preferable converts terpene moiety to a 6-member ring. TleD was the only bMTC identified that fitted all the criteria when we conceived the idea in early 2018. The other possible bMTC that takes terpene moiety is sodC. However, it forms 5-member ring from farnesyl pyrophosphate and the enzyme was only reported in late 2018⁷. Thus, we didn't include this enzyme in our initial screening. In addition, we have tested other methyltransferases that only partially fulfil the criteria, as shown in the table R1 below. However, none of them converted psi-ionone to irone. We have modified the first paragraph of the results section to include the information and table R1 has been added as the new supplementary table S1.

Table R1. Selected methyltransferases with known structures were tested for their potential activity for conversion of psi-ionone to irone.

Enzyme	Protein name	PDB ID	Methylation	Cyclization	Natural substrate contains terpene moiety?	Forms 6-member ring?	TTN
Cyclopropane mycolic acid synthase 1	Mt.CMAS	1KPG	Yes	Yes	No	No	ND
Geranyl diphosphate 2-C-methyltransferase	Sl.GdpMT	4F86	Yes	No	Yes	No	ND
Methyltransferase-like protein	Ss.SpnF	4PNE	No	Yes	No	No	ND
O-methyltransferase	TleD (pMT1)	5GM2	Yes	Yes	Yes	Yes	<0.001

3. In lines 49-57, the authors report part of their results already in the introduction. The manuscript must be reorganized so that these results become the first paragraph of the Results section.

We have reorganized the introduction and the first paragraph of the results section (Line 77-84, highlighted in red text) accordingly so that the results are reported in the results section instead of in the introduction.

4. Figure 2 contains panels with limited information (such as panel a), while Fig S2 in the supplementary information contains important detailed information that should be part of the main manuscript.

We have swapped the Fig. 2a and Fig. S2a and they are the new Fig. S1a and Fig. 2a, respectively.

5. How does the mechanism of cyclization by TleD in teleocidin synthesis compare with the mechanism of converting psi-ionone to irones? Can the authors provide a comparison of the catalytic steps and draw analogies /similarities?

We do not have the experimental evidence for the mechanism and the following mechanism is proposed based on literature information. Figure R4a shows the proposed cyclization mechanism based on Yu. et al⁸. Figure R4b shows the proposed cyclization mechanism based

on Marner et. al⁹. The proposed teleocidin synthesis involves the formation of 5-member ring first and expands to 6-member ring. However, for the irone synthesis, 6-member ring is directly formed. This might be attributed to the difference in the conjugated double bond between the starting substrate, namely teleocidin A1 and psi-ionone.

Figure R4. The proposed mechanisms for TleD in a) teleocidin synthesis and b) irone synthesis.

6. Reference 1 is hardly relevant for the argument used. The authors must consider references that clearly describe the field and not only cite their work.

We have added a few more references (line 33, highlight in red) which have shown nearly theoretical production of the molecules.

7. Lines 24-25. The authors term this approach integrated retrobiosynthetic pathway design. This may be exaggerated considering that the discovery of pMT1, was, according to the authors, serendipitous.

We thank the reviewer's comment. However, retrobiosynthetic pathway design and the choice of enzyme are two distinct concepts. Retrobiosynthetic pathway design is walking backwards from a target molecule and using the biotransformation rules to reconstruct a biochemical pathway¹⁰. This design is a putative biochemical pathway that suggests the biotransformation rules but does not identify the right enzyme. Thus, we have designed a retrosynthetic pathway to produce irone. To achieve that, we are looking for a bifunctional methyltransferase and cyclase (biotransformation rule) that is preferably converting terpene moiety to a 6-member ring. As mentioned in our response to reviewer 3 comment 2, the discovery of the pMT1 was serendipitous, as it was the only known enzyme that fulfilled all the criteria. We have modified the first paragraph of the results section to explain our design concept and include screening results of various methyltransferases (line 77-84 and line 91-95, text highlighted in red).

8. Line 70 and in several other places in the main text and SI. The authors use the term "directed evolution" when describing their approach. In my opinion, the approach is based on rational design and I do not see basic elements of directed evolution being implemented here. The use of saturation mutagenesis is common to many protein engineering approaches.

We thank the reviewer's comment. We have modified the text to change the "directed evolution" to "structure-guided rational design" (the modified text is highlighted in red).

9. What is the second band in the western blot in figure S4? A fragmentation product? Why is it not present in the lanes with the wild-type enzyme?

We thank the reviewer's comment, and we agree that it may be a fragment product. However, we did not investigate the second band in the western blot as it is beyond the aim of the study. We have purified all the mutants and characterized them again with purified protein as suggested by the reviewer (the new supplementary figure S3b). The western blot data will not be used to quantify the protein concentration but to indicate relative soluble amount of each mutant found in the supernatant. Therefore, we did not look into the second band in the western blot.

10. If the lanes in Fig S2 are all equally loaded, then do the different levels of the different indicate? Is this variation due to protein stability or is the variation in gene expression so large between samples? As discussed above, it is essential to isolate and characterize all these variants using *in vitro* assays to be able to make meaningful conclusions about the results.

We believe the reviewer is referring to the original Fig. S4a. The loading is based on the same amount of cells, namely $40 \times 600 \text{ OD} \times \mu\text{l}$ cells. All the genes were cloned into pET vector and under T7 promoter. Thus, the gene expression between the samples is believed to be similar. The difference is likely because of protein solubility, as only the soluble fraction of the protein is loaded. We have purified all the mutant proteins and characterize them *in vitro*. The results is shown as the new supplementary figure S3b.

Reference

1. Chen, X., Shukal, S. & Zhang, C. Integrating Enzyme and Metabolic Engineering Tools for Enhanced α -Ionone Production. *J. Agric. Food Chem.* (2019)
doi:10.1021/acs.jafc.9b00860.
2. Newman, J. D. *et al.* High-level production of amorpho-4,11-diene in a two-phase partitioning bioreactor of metabolically engineered *Escherichia coli*. *Biotechnology and Bioengineering* **95**, 684–691 (2006).
3. Alonso-Gutierrez, J. *et al.* Metabolic engineering of *Escherichia coli* for limonene and perillyl alcohol production. *Metabolic Engineering* **19**, 33–41 (2013).
4. Effects of hydrocarbon additions on gas-liquid mass transfer coefficients in biphasic bioreactors | SpringerLink. <https://link-springer-com.ejproxy.a-star.edu.sg/article/10.1007/BF02932038>.

5. Zhang, R. K. *et al.* Enzymatic assembly of carbon–carbon bonds via iron-catalysed sp³ C–H functionalization. *Nature* **565**, 67–72 (2019).
6. Tufvesson, P., Lima-Ramos, J., Nordblad, M. & Woodley, J. M. Guidelines and Cost Analysis for Catalyst Production in Biocatalytic Processes. *Org. Process Res. Dev.* **15**, 266–274 (2011).
7. von Reuss, S. *et al.* Sodorifen Biosynthesis in the Rhizobacterium *Serratia plymuthica* Involves Methylation and Cyclization of MEP-Derived Farnesyl Pyrophosphate by a SAM-Dependent C-Methyltransferase. *J. Am. Chem. Soc.* **140**, 11855–11862 (2018).
8. Yu, F. *et al.* Crystal structure and enantioselectivity of terpene cyclization in SAM-dependent methyltransferase TleD. *Biochem. J.* **473**, 4385–4397 (2016).
9. Marner, F.-J., Gladtko, D. & Jaenicke, L. Studies on the Biosynthesis of Iridals and Cycloiridals. *Helvetica Chimica Acta* **71**, 1331–1338 (1988).
10. Hadadi, N. & Hatzimanikatis, V. Design of computational retrobiosynthesis tools for the design of de novo synthetic pathways. *Current Opinion in Chemical Biology* **28**, 99–104 (2015).

REVIEWER COMMENTS

Reviewer #1 (Remarks to the Author):

The authors have revised the paper according to the reviewers suggestions. The paper can be published as it is.

Reviewer #2 (Remarks to the Author):

The authors have addressed my issues. I have no other questions and suggest to accept this manuscript.

Reviewer #3 (Remarks to the Author):

The authors have satisfactorily answered my concerns. Congratulations on this very interesting piece of work.

REVIEWERS' COMMENTS

Reviewer #1 (Remarks to the Author):

The authors have revised the paper according to the reviewers suggestions. The paper can be published as it is.

We would like to thank the reviewer for your valuable suggestions and positive comments.

Reviewer #2 (Remarks to the Author):

The authors have addressed my issues. I have no other questions and suggest to accept this manuscript.

We would like to thank the reviewer for your valuable suggestions and positive comments.

Reviewer #3 (Remarks to the Author):

The authors have satisfactorily answered my concerns. Congratulations on this very interesting piece of work.

We would like to thank the reviewer for your valuable suggestions and positive comments.